# Anithiactin D, a Phenylthiazole Natural Product from Mudflat-Derived *Streptomyces* sp., Suppresses Motility of Cancer Cells

**DOI:** 10.3390/md22020088

**Published:** 2024-02-14

**Authors:** Sultan Pulat, Inho Yang, Jihye Lee, Sunghoon Hwang, Rui Zhou, Chathurika D. B. Gamage, Mücahit Varlı, İsa Taş, Yi Yang, So-Yeon Park, Ahreum Hong, Jeong-Hyeon Kim, Dong-Chan Oh, Hangun Kim, Sang-Jip Nam, Heonjoong Kang

**Affiliations:** 1College of Pharmacy, Sunchon National University, Sunchon 57922, Republic of Korea; sultanpulat@s.scnu.ac.kr (S.P.); zhourui274@gmail.com (R.Z.); chathurika.gamage@gmail.com (C.D.B.G.); mucahitvarli@s.scnu.ac.kr (M.V.); mr.isatas@gmail.com (İ.T.); yangyi_520@hotmail.com (Y.Y.); sinbu17@naver.com (S.-Y.P.); 2Department of Convergence Study on the Ocean Science and Technology, Korea Maritime and Ocean University, Busan 49112, Republic of Korea; ihyang@kmou.ac.kr; 3Laboratories of Marine New Drugs, REDONE Seoul, Seoul 08594, Republic of Korea; ljh@urc.kr; 4Department of Chemistry and Nanoscience, Ewha Womans University, Seoul 03760, Republic of Korea; har@urc.kr (A.H.); sub03101@ewhain.net (J.-H.K.); 5Natural Products Research Institute, College of Pharmacy, Seoul National University, NS-80, Seoul 08826, Republic of Korea; sunghooi@snu.ac.kr (S.H.); dongchanoh@snu.ac.kr (D.-C.O.); 6Laboratory of Marine Drugs, School of Earth and Environmental Sciences, Seoul National University, NS-80, Seoul 08826, Republic of Korea

**Keywords:** marine natural products, gastric cancer, colorectal cancer, lung cancer, EMT, MMP, Rho GTPases

## Abstract

Anithiactin D (**1**), a 2-phenylthiazole class of natural products, was isolated from marine mudflat-derived actinomycetes *Streptomyces* sp. 10A085. The chemical structure of **1** was elucidated based on the interpretation of NMR and MS data. The absolute configuration of **1** was determined by comparing the experimental and calculated electronic circular dichroism (ECD) spectral data. Anithiactin D (**1**) significantly decreased cancer cell migration and invasion activities at a concentration of 5 μM via downregulation of the epithelial-to-mesenchymal transition (EMT) markers in A549, AGS, and Caco-2 cell lines. Moreover, **1** inhibited the activity of Rho GTPases, including Rac1 and RhoA in the A549 cell line, suppressed RhoA in AGS and Caco-2 cell lines, and decreased the mRNA expression levels of some matrix metalloproteinases (MMPs) in AGS and Caco-2 cell lines. Thus **1**, which is a new entity of the 2-phenylthiazole class of natural products with a unique aniline-indole fused moiety, is a potent inhibitor of the motility of cancer cells.

## 1. Introduction

Cancer affects one in five people worldwide, making it one of the world’s most serious public health challenges [1]. Metastatic progression and resistance to therapy contribute to fatal outcomes [2]. Cancer that has spread beyond its point of origin to a distant part of the body is called metastatic cancer. Epithelial-to-mesenchymal transition (EMT), matrix metalloproteinases (MMPs), and Ras superfamily of small GTPases including Rac1 and RhoA are involved in cancer cell motility that is associated with signaling pathways. During EMT, epithelial cells become mesenchymal, and cancer cells can migrate to other parts of the body, thus contributing to the spread of cancer [3]. MMPs are correlated with cancer aggressiveness when overexpressed and high enzymatic activity [4]. Ras homologous GTPases play a crucial role by influencing actin reorganization and cytoskeletal arrangement in cancer metastasis and invasion [5]. Therefore, suppressing EMT, MMPs, and Rho GTPases is important for developing effective cancer therapies.

In 2014, anithiactins A–C were isolated from *Streptomyces* sp. 10A085 derived from marine mudflat sediments. They were the first 2-phenylthiazole natural products bearing an aniline moiety to be reported [6]. Shortly thereafter, a marine-derived bacterium, *Actinomycetospora* sp., has also produced anithiactins A and C along with thiasporine A [7,8]. Several studies regarding the synthesis of these secondary metabolites have been reported [6,9,10,11]. Intensive investigation of the chemical components of the culture broth of the strain 10A085 has led to the isolation of a new 2-phenylthiazole natural product, anithiactin D (**1**). Herein, we describe the isolation, structure elucidation, and effect of **1** on cancer cell motility.



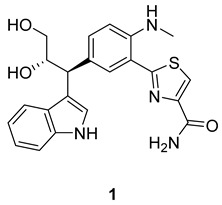



## 2. Results

### Structure Elucidation

Anithiactin D (**1**) was isolated as a yellowish amorphous powder, and its molecular formula was determined as C_22_H_22_N_4_O_3_S based on HRFABMS. The ^1^H NMR spectrum of 1 revealed a 1, 2-disubstituted benzene [H-6″ (δ_H_ 7.32, d, J = 7.8 Hz), H-7″ (δ_H_ 7.04, dd, J = 7.8, 7.8 Hz), H-8″ (δ_H_ 6.94, dd, J = 7.8, 7.8 Hz), and H-9″ (δ_H_ 7.48, d, J = 7.8 Hz)] and a 1,2,4-trisubstituted benzene ring moieties[H-3 (δ_H_ 6.74, d, J = 7.8 Hz), H-4 (δ_H_ 7.42, dd, J = 7.8, 1.3 Hz), and H-6 (δ_H_ 7.76, d, J = 1.3 Hz)]. The ^1^H NMR spectrum of **1** also displayed two downfield singlet protons [H-4′ (δ_H_ 8.07, s), H-2″ (δ_H_ 7.27, s)] and one N-methyl singlet [2-NHMe (δ_H_ 2.96, s)], two deshielded methines [H-10″ (δ_H_ 4.27, d, J = 7.0 Hz) and H-11″ (δ_H_ 4.41, ddd, J = 7.0, 7.0, 4.2 Hz)] and one methylene [H-12″a (δ_H_ 3.61, dd, J = 7.0, 4.2 Hz) and H-12″b (δ_H_ 3.48, dd, J = 7.0, 4.2 Hz)]. The presence of a propane-1,2-diol moiety was inferred from the COSY crosspeaks [H-10″/H-11/H-12] and their carbon chemical shifts [C-10″ (δ_C_ 45.6), C-11″ (δ_C_ 75.6), C-12″ (δ_C_ 66.2)]. The HMBC correlations from singlet proton H-4′ to carbons C-2′ (δ_C_ 171.7), C-5′ (δ_C_ 150.6) and C-6′ (δ_C_ 165.6) indicated a thiazole ring moiety. The HMBC correlations from H-6 to carbons C-2 (δ_C_ 147.0), C-4 (δ_C_ 134.2), C-5 (δ_C_ 130.8) and C-2′ allowed the connection of the thiazole ring to the 1,2,4-trisubstituted benzene ring. The N-methyl singlet was placed at C-2 based on the observation of the three-bond HMBC correlation from the 2-NHMe to C-2. An indole moiety was deduced from the HMBC correlations from singlet proton H-2″ (δ_H_ 7.27, s) to carbons C-3″ (δ_C_ 117.8), C-4″ (δ_C_ 128.2) and C-5″ (δ_C_ 137.9). The attachment of C-3″/C-10″/C-5 was deduced from the three-bond HMBC correlations of H-2″ to C-10″ as well as the observation of HMBC correlations from the methine proton H-10″ to carbons C-4, C-5, C-6, C-3″, and C-4″. Thus, the planar structure of **1** was assigned as shown in Figure 1.

The absolute configurations at C-10″ and C-11″ in **1** were determined as 10″*S* and 11″*S*, respectively, by a comparison of experimental and calculated electronic circular dichroism (ECD). The experimental ECD spectrum of **1** had distinct positive Cotton effects at 229 nm and 249 nm and negative Cotton effects at 290 nm and 390 nm (Figure 2). The comparison of calculated ECD spectra of four stereoisomers for two stereocenters at C-10″ and C-11″ using the density functional theory (DFT) model showed that only one theoretical isomer with 10″*S* and 11″*S* configurations was in accordance with the experimental ECD spectrum of **1** (Figure 2 and Appendix A).

The chemical structure of **1** shares a core aniline–thiazole unit with other anithiactins. Aeruginoic acid was considered to be a biosynthesis shunt during the pathway of pyochelin, originating from a cysteine unit [12]. A cascade amination procedure would result in the formation of anithiactin B. Compound **1** might result from a conjugation reaction between anithiactin B and indole-3-glycerol (Figure 1). While indole-3-glycerol phosphate is mostly suggested as a participating building block in the biosynthetic pathway, connecting through the α-carbon or carboxy carbon, **1** features a β-carbon-linked product, implying the presence of an unusual biosynthetic pathway. The stereochemistry observed in **1** supports the proposed biosynthetic pathway.

Indole has been recognized as a crucial pharmacophore in bioactive natural products [13]. Extensive research has been conducted to incorporate an indole moiety into bioorganic compounds, employing methods ranging from organic synthesis to biosynthetic approaches [14,15]. The indole moiety within natural products is commonly derived from tryptophan. The formation of C-C bond attachments of the indole group with other entities typically occurs through the α-carbon or carboxylic acid carbon, resulting in the chemical characteristics of an amino acid unit. As a result, there are few studies of natural products containing tryptophan with a β-carbon linkage. Discoipyrrole D was reported in 2013, with a polycyclic pyrrole core [16]. The suggested biosynthetic pathway indicates a post-mortem attachment of the tryptophan moiety. Cytotoblastin possesses an indolactam V skeleton with an additional tryptophan unit linked through the β-carbon [17]. Notably, both the tryptophan β-carbon-linked moiety bearing natural products exhibited inhibitory effects on cancerous cells.

The biosynthetic pathways of phenyl-thiazoline and -thiazole motifs have been intensely studied due to their crucial roles as metal chelators and bioactive pharmacophores [18]. In 2019, pyonitins were reported along with their corresponding biosynthetic gene clusters. These compounds were the outcome of a non-enzymatic reaction between pyochelin and pyrrolnitrin [19]. However, all phenyl-thiazoline and -thiazole natural products showed a linkage to the main chemical framework via the C4 position of the pentacycle, resulting in the phenyl as a side functional group. Compound **1** is the first example of a para-extended phenyl-thiazole natural product.

Compound **1** was investigated regarding its cytotoxicity against a variety of cancer cell lines. Relative cell viabilities of A549, AGS, and Caco-2 cells were measured by MTT assay after treatment with various concentrations (10, 25, 50, and 100 μM) of **1** for 48 h. Treatment with 100 μM of **1** did not affect the cell viabilities of A549 and Caco-2 cells, whereas it decreased the cell viability of AGS cells (Figure 3). These results indicate that among the three cells, **1** is slightly associated with reduced cell viability of AGS.

During metastasis, cancer cells migrate away from the primary tumor via a process known as tumor invasion [20]. Metastasis occurs when tumor cells spread from the primary site to the rest of the body, causing severe organ damage and death [21]. Migration and invasion assays using AGS, A549, and Caco-2 cells were performed to determine whether **1** affects cancer cell motility. Migration of AGS, A549, and Caco-2 cells was inhibited by 5 μM of **1** by ~35%, ~35%, and ~45%, respectively (Figure 4A,B), and invasion of these three cell lines was inhibited by ~30%, ~50%, and ~50%, respectively, more than were the DMSO-treated cells (Figure 4C,D). Overall, these results indicate that nontoxic concentrations of **1** significantly suppress cancer cell motility.

Cancer cells migrate across neighboring cells and tissues via EMT, wherein polarized epithelial cells are transformed into motile mesenchymal cells, which is a primary factor for the morphological changes in various physiological processes [22]. We examined the mRNA expression levels of EMT effectors and EMT transcription factors in A549, AGS, and Caco-2 cells to determine whether **1** inhibited motility. The mRNA expression of E-cadherin was increased by 5 μM of **1**, whereas that of N-cadherin and EMT transcription factors (Snail, Slug, Twist, Zeb1, and Zeb2) were decreased in AGS and A549 cells (Figure 5A,B). The mRNA expression of E-cadherin was not affected by 5 μM of **1** while that of N-cadherin and these EMT transcription factors were decreased in Caco-2 cells (Figure 5C). These results show that **1** decreases N-cadherin expression by downregulating EMT transcription factors.

Cell proliferation, gene expression, and membrane trafficking are all regulated by Rho GTPases [23]. Hence, assays to analyze the affinity-precipitation of cellular GTPases were performed to determine whether treatment with **1** at nontoxic concentrations affects the motility of AGS, A549, and Caco-2 cells. These tests showed that 5 μM of **1** significantly decreased the activities of RhoA and Rac1 in A549 cells (Figure 6A,B). Moreover, RhoA activity reduced, and Rac1 activity was not affected by treatment with **1** in AGS and Caco-2 cells (Figure 6C,D). These results demonstrate that **1** inhibits cancer cell motility by downregulating the activity of Rho GTPases.

The important collagen- and elastin-binding properties of MMPs help their binding to extracellular matrix (ECM) components. MMP-2 and MMP-9 play a significant role in cytotrophoblast invasion of maternal vessels [24]. The effect of **1** on the mRNA expression levels of MMP2 and MMP9 in AGS, A549, and Caco-2 was examined using qRT-PCR. Treatment with 5 μM of **1** significantly decreased the mRNA expression levels of MMP2 in AGS cells, while it significantly decreased those of MMP2 and MMP9 in Caco-2 cells (Figure 7). Additionally, **1** did not affect the mRNA expression levels of MMP2 and MMP9 in A549 cells (Figure 7). Subsequently, qPCR assays were conducted to measure the mRNA expression levels of TIMP molecules, such as TIMP1 and TIMP2. Matrix MMP inhibitors (TIMP1 and TIMP2) mediate cell proliferation, migration, and invasion [25]. The mRNA expression levels of TIMP1 and TIMP2 were not affected by treatment with **1** (Figure 7). 

Based on statistics, lung cancer is the first, colorectal cancer is the second, while gastric cancer is the fourth most common cause of cancer death [26]. In this study, we selected three different cancer cell lines, which are A549 (lung cancer), AGS (gastric cancer), and Caco-2 (colorectal cancer) for MTT assay. Here, we found that **1** shows potent activity in decreasing the cell viability of AGS compared with A549 and Caco-2.

The induction of EMT leads to cancer metastasis via EMT transcription factors including Snail, Slug, Twist, Zeb1, and Zeb2. The upregulation of EMT is one of the reasons for therapy resistance in tumor cells [27]. Our results show that **1** modulates the expression of the EMT effectors by downregulating the transcription factors, Snail, Slug, Twist, Zeb1, and Zeb2.

In cancer metastasis and invasion, Ras homologous GTPases play an important role by influencing actin reorganization and cytoskeletal arrangement. Some types of tumors are associated with carcinogenesis and the progression of Rho GTPases, such as RhoA and Rac1 [23]. By activating p21-activated kinases, Rac1 is crucial for regulating actin cytoskeleton and promoting cell growth [28]. Our Western blot assays show that **1** suppresses cancer cell motility by downregulating the activity of Rho GTPases.

MMPs play an important role from the initial stages of cancer development to the establishment of a metastatic niche in another organ by breaking down some components of ECM [29]. All the main components of the ECM and basement membrane are degraded by the 24 MMPs. In the early stages of cancer progression, MMPs are involved in the destruction of tumor tissue and development of metastasis [30]. This study demonstrates that **1** decreases the mRNA expression level of some MMPs in AGS and Caco-2 cell lines, but does not affect MMPs in A549. 

TIMP molecules, including TIMP1 and TIMP2, reduce the activities of MMPs; therefore, they can control the remodeling of the extracellular matrix in line with the interaction of cells. However, the mRNA expression levels of TIMP1 and TIMP2 were not affected by treatment with **1**. The present findings show that TIMP1 and TIMP2 do not play a role in the MMP-reducing effect of **1**.

## 3. Materials and Methods

### 3.1. General Experimental Procedures

The optical rotation was measured using the Autopol III polarimeter with a 5 cm cell. Ultraviolet (UV) spectra were recorded on the Scinco UVS-2100 spectrophotometer and infrared (IR) spectra were recorded on the Scimitar 800 FT-IR spectrometer. Nuclear magnetic resonance (NMR) spectra were obtained using the Bruker Avance 700 spectrometer. Methanol (MeOH; δ_H_ 3.31, δ_C_ 49.3) resonances were used as internal references. High resolution fast atom bombardment mass spectroscopy (HRFABMS) was performed using the JEOL JMS-600W spectrometer (JEOL Ltd. Tokyo, Japan). Low-resolution LC/MS measurements were performed using the Agilent Technologies 1260 quadrupole and Waters Micromass ZQ LC/MS system using a reversed-phase column (Phenomenex Luna C18 (2) 100 Å, 50 mm × 4.6 mm, 5 µm) at a flow rate of 1.0 mL/min at the National Research Facilities and Equipment Center (NanoBioEnergy Materials Center) at Ewha Womans University. Column chromatography separation was performed using a C18 column via eluting with a gradient of MeOH and water (H_2_O). The fractions were purified using the Waters 1525 binary high-performance liquid chromatography (HPLC) pump with a reversed-phase C18 column (Phenomenex Luna C18 (2), 250 mm × 10 mm, 5 μm) via eluting with 30% acetonitrile (CH_3_CN) in H_2_O at flow rate of 2.0 mL/min.

### 3.2. Strain Isolation

*Streptomyces* sp. 10A085 was isolated from marine sediments collected from Jaebu Island, Gyeonggi-Do, South Korea in 2010. The mud sediment samples were dried in air for 24 h on a clean bench and subjected to heat shock at 55 °C for 5 min to eliminate other bacteria. Aggregated clumps were gently dispersed using a glass rod and then placed onto ISP medium 1 and ISP medium 4 solid agar substrates using a sponge plug. Additionally, some dried samples were rehydrated in sterilized seawater, and the resulting diluted suspension was spread on the solid agar substrates using a plastic rod. These crude plates were placed in a chamber maintained at 27 °C and monitored for 1–3 months to obtain unique actinomycete-like colonies. Strain 10A085 was picked from an ISP medium 4 agar plate containing white spores. The 16S rRNA gene that was cloned using universal primers 27F and 1492R showed 99.5% (1350/1357) similarity to *Streptomyces* sp. HV10 (accession no. KM881709).

### 3.3. Fermentation, Extraction, and Purification

Strain 10A085 was cultured at 27 °C via shaking at 130 rpm in 70 × 2.5 L Ultra Yield Flasks, each containing 1 L of yeast malt extract (YME) medium (4 g of yeast extract, 10 g of malt extract, and 2 g of glucose mixed with 34.75 g of artificial sea salt and dissolved in distilled H_2_O). After 13 days, the broth was extracted using ethyl acetate (EtOAc), and the solvent was dried in vacuo to yield 2.6 g of extract. It was separated by medium-pressure liquid chromatography (MPLC) using a silica column and step-gradient elution (0%, 1%, 2%, 5%, 10%, 20%, 50%, 90%, and 100%) of MeOH in dichloromethane (CH_2_Cl_2_). Fraction 3 was further purified by reversed-phase HPLC under isocratic conditions using 35% aqueous CH3CN to obtain 3.5 mg of **1** [31]. 

Anithiactin D (1): amorphous yellowish powder; [α]^25^_D_ +121 (c 0.042, MeOH); UV (MeOH) λ_max_ (log ε) 220 (2.15) and 230 (2.09) nm; IR (KBr) ν_max_ 3435 and 1646 cm^−1^; ^1^H and ^13^C NMR data, see Table 1; (+)-HRESIMS, *m/z* 423.1483 [M + H]^+^ (calculated for C_22_H_23_N_4_O_3_S, 423.1491).

### 3.4. ECD Calculation

The conformational search of **1** for ECD calculation was conducted using MacroModel employing the Merck molecular force field in the gas phase. To minimize computational complexity and expense, an upper energy limit of 10 kJ/mol and a convergence threshold of 0.001 kJ (mol Å)^−1^ on the rms gradient were employed. The Boltzmann population of each conformer was calculated based on this condition. To calculate ECD spectra corresponding to the optimized structures, DFT calculations were performed using the B3LYP/DFT functional level and def-SV(P) basis set by Turbomole X 4.3.2. The ECD spectra were simulated by overlapping each transition, where the width of the band at 1/e height (*σ*) was set to 0.01 eV. The excitation energies (Δ*E_i_*) and rotary strengths (*R_i_*) for transition (*i*) were considered during the simulation. 

### 3.5. Cell Culture

Three different cancer cell lines, A549 (lung cancer), AGS (gastric cancer), and Caco-2 (colorectal cancer), were cultured in Roswell Park Memorial Institute (RPMI) medium or Dulbecco’s modified Eagle’s medium (DMEM; GenDepot, Katy, TX, USA). These were supplemented with 10% fetal bovine serum (FBS) and 1% penicillin–streptomycin (GenDepot) solution in a humidified atmosphere of 5% carbon dioxide (CO_2_) at 37 °C [32]. 

### 3.6. MTT Assay

The viabilities of A549, AGS, and Caco-2 were tested using the MTT assay (Sigma-Aldrich, St. Louis, MO, USA). AGS (2 × 10^3^ cells/well), A549 (2 × 10^3^ cells/well), and Caco-2 (3 × 10^3^ cells/well) cells were seeded on a 96-well plate. The cancer cells were then treated with DMSO (Sigma-Aldrich) or different concentrations of **1** and incubated for 48 h and then with MTT for 4 h [33].

### 3.7. Cell Migration and Invasion Assays

Invasion assays were performed in chambers (Corning, New York, NY, USA) containing polycarbonate membranes having 8 μm pores coated with 1% gelatin and migration assays in chambers containing polycarbonate membranes with 8 μm pores without coating. Cells (2–3 × 10^5^) in 120 µL of medium (DMEM/RPMI 1640 containing 0.2% bovine serum albumin (BSA) dissolved in phosphate-buffered saline (PBS) with or without **1**) were seeded in the upper chamber. The cells in the upper chamber were fixed using the Diff Quik kit (Sysmex, Kobe, Japan) after 24 h of incubation. An upright microscope was used to count the cells adhering to the underside of the chamber [34].

### 3.8. Quantitative Real-Time PCR

Cells (2.5 × 10^5^) were seeded in six-well plates (Corning) and treated with 1, 2.5, and 5 μM of **1**, incubated at 37 °C for 24 h. RNAiso Plus (TaKaRa, Siga, Japan) was used to isolate total RNA from A549, AGS, and Caco-2 cells for qRT-PCR. Total RNA was converted into cDNA using the Moloney murine leukemia virus (M-MLV) reverse transcriptase kit (Invitrogen, Carlsbad, CA, USA), and SYBR Green reagent (Enzynomics, Seoul, Republic of Korea) was used to determine the relative expressions of candidate gen [35].

### 3.9. Affinity-Precipitation of Cellular GTPases

As previously described, GST-RBD/PBD was used to determine the RhoA and Rac1 activity in cells [36,37]. We incubated the lysates at room temperature for 1 h with GST-RBD/PBD beads which were then washed four times with washing buffer. Monoclonal antibodies against RhoA and Rac1 were detected by immunoblotting. Multi-Gauge software, version 3.0 was used to determine the relative activities of each GTPase by quantifying each band of GTP-bound GTPase. The GTP-bound bands were normalized to the total amount.

## 4. Conclusions

Anithiactin D (**1**), a phenylthiazole secondary metabolite, was isolated from marine-derived *Streptomyces* sp. 10A085. We discovered that **1** downregulates the transcription factors, Sail, Slug, Twist, Zeb1, and Zeb2, which in turn modulate the mRNA expression levels of EMT effectors, namely N-cadherin and E-cadherin. The activities of Rho GTPases were also decreased by **1**. In addition, **1** suppressed the mRNA expression levels of some MMPs (Figure 8). These results demonstrate that 1 is a potent inhibitor of the motility of cancer cells.

## Data Availability

The data presented in this study are available in the Appendix A.

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
