# Peer review of "Anithiactin D, a Phenylthiazole Natural Product from Mudflat-Derived Streptomyces sp., Suppresses Motility of Cancer Cells"

_marinedrugs, 2024, doi:10.3390/md22020088_

Round 1
Reviewer 1 Report
Comments and Suggestions for Authors
The authors of the present manuscript (marinedrugs-2849349) report the isolation-structure elucidation of a new anithiactin from a mudflat- derived Streptomyces sp. The authors also determined the absolute configuration of the new metabolite by comparison of the experimental and calculated electronic circular dichroism spectra. Additionally, the authors evaluated the anticancer activity of anithiactin D and showed that the new 2-phenylthiazole is a potent inhibitor of the motility of cancer cells.
The manuscript is well-structured and the structure elucidation/absolute configuration determination is clearly presented and based on safe interpretation of the obtained spectroscopic data.
Concerning the activity evaluation, I cannot provide a very comprehensive opinion since it is not my field of expertise. However, the study seems well organized and in-depth, revealing the potential of the newly discover compound.
Minor changes include:
Figure 8 should be moved from the Conclusions to the Results and Discussion Section.
Line 69: The ….two down-fielded methines…. should be replaced by …..two deshielded methines…..
In several places the number 1 referring to anithiactin D is not in bold.
The refences should include the title of the articles and DOI.
Comments on the Quality of English Language
Minor editing is required.
Reviewer 2 Report
Comments and Suggestions for Authors
The manuscript is generally well-written and organized, and may be acceptable for publication in Marine Drug after minor revisions.
Please address the following issues.
1) Line 30: in the A549 cell line → in A549 cell line
2) Line 53: Please check the phrase “has also been produced”. Is “been” necessary?
3) Lines 57-58: The term “anti-cancer activities” is typically used in the context of in vivo evaluations. As this is an in vitro study, please consider revising the expression.
4) Lines 79-80: A long-range correlation from H-2’’ to C-6’’ is not displayed in Figure 1. Was a 4JC-H correlation observed?
5) Line 82: C-4, C-5, C-6, C-3’’, C-4’’ → C-4, C-5, C-6, C-3’’ and C-4’’
6) Line 84: 10’’S, 11’’S → 10’’S and 11’’S, respectively,
7) Line 100: ani-line-thiazole → aniline-thiazole
8) Line 103: 1 might result → Compound 1 might result
9) Line 106: carboxyl → carboxy
10) Line 131: 1 is the first → Compound 1 is the first
11) Line 133: 1 was performed for → Compound 1 was performed for
12) Lines 137-138: The authors describe “These results indicate that 1 is highly associated with reduced cell viability of AGS.” However, since 1 only reduces the cell viability of AGS by 20% at a high sample concentration of 100 μM, the cytotoxicity of 1 is considered to be weak. Therefore, the authors may need to reconsider the phrasing.
13) Lines 141, 149, 280 and 286: The compound number 1 should be changed in bold-font style.
14) In Figure 4, Figure 5 and Figure 6, (a), (b), (c) and (d) should be revised as (A), (B), (C) and (D).
15) Lines 182 and 219: Please remove “compound” from “compound 1”.
16) Line 300: fetal bovine serum → fetal bovine serum (FBS)
17) Lines 303-304: (Sigma-Aldrich, St. Louis, USA) → (Sigma-Aldrich, St. Louis, MO, USA)
18) Line 306: (Sigma-Aldrich, St. Louis, USA) → (Sigma-Aldrich)
19) Line 309: (Corning, New York, USA) → (Corning, New York, NY, USA)
20) Line 315: (Sysmex Kobe, Japan) → (Sysmex Hyogo, Japan)
21) Line 318: (Corning, New York, USA) → (Corning)
22) Line 319: (TaKaRa, Otsu, Japan) → (TaKaRa, Siga, Japan)
23) Line 323: (Invitrogen, Carlsbad, USA) → (Invitrogen, Carlsbad, CA, USA)
Comments on the Quality of English Language
The English text is well written and there are no particular problems.
